# Deep Learning-Based Segmentation of 3D Volumetric Image and Microstructural Analysis

**DOI:** 10.3390/s23052640

**Published:** 2023-02-27

**Authors:** Bahar Uddin Mahmud, Guan Yue Hong, Abdullah Al Mamun, Em Poh Ping, Qingliu Wu

**Affiliations:** 1Department of Computer Science, Western Michigan University, Kalamazoo, MI 49008, USA; 2Faculty Engineering and Technology, Multimedia University, Melaka 75450, Malaysia; 3Department of Chemical and Paper Engineering, Western Michigan University, Kalamazoo, MI 49008, USA

**Keywords:** image segmentation, 3D image, 3D UNET-VGG19, image microstructure, particle analysis, image reconstruction

## Abstract

As a fundamental but difficult topic in computer vision, 3D object segmentation has various applications in medical image analysis, autonomous vehicles, robotics, virtual reality, lithium battery image analysis, etc. In the past, 3D segmentation was performed using hand-made features and design techniques, but these techniques could not generalize to vast amounts of data or reach acceptable accuracy. Deep learning techniques have lately emerged as the preferred method for 3D segmentation jobs as a result of their extraordinary performance in 2D computer vision. Our proposed method used a CNN-based architecture called 3D UNET, which is inspired by the famous 2D UNET that has been used to segment volumetric image data. To see the internal changes of composite materials, for instance, in a lithium battery image, it is necessary to see the flow of different materials and follow the directions analyzing the inside properties. In this paper, a combination of 3D UNET and VGG19 has been used to conduct a multiclass segmentation of publicly available sandstone datasets to analyze their microstructures using image data based on four different objects in the samples of volumetric data. In our image sample, there are a total of 448 2D images, which are then aggregated as one 3D volume to examine the 3D volumetric data. The solution involves the segmentation of each object in the volume data and further analysis of each object to find its average size, area percentage, total area, etc. The open-source image processing package IMAGEJ is used for further analysis of individual particles. In this study, it was demonstrated that convolutional neural networks can be trained to recognize sandstone microstructure traits with an accuracy of 96.78% and an IOU of 91.12%. According to our knowledge, many prior works have applied 3D UNET for segmentation, but very few papers extend it further to show the details of particles in the sample. The proposed solution offers a computational insight for real-time implementation and is discovered to be superior to the current state-of-the-art methods. The result has importance for the creation of an approximately similar model for the microstructural analysis of volumetric data.

## 1. Introduction

This research builds upon our previous work [1] that leveraged a CNN-based 2D UNET architecture for multiclass 2D image segmentation. In this paper, we proposed using 3D UNET in combination with VGG19 for modeling and multiclass segmentation of volumetric images as well as subsequent particle-level analysis. With deep learning, we can recognize objects with unprecedented precision. Object detection [2], deepfake technology [3], and human–AI collaborative development [4] are just a few examples of the many fields where deep learning models are employed. All of the above-mentioned works somehow involved experimentation with image data. The majority of techniques are computationally expensive and labor-intensive, rendering them unsuitable for real-time applications such as robotics, virtual reality, and medical imaging. Another concern is that many current techniques have memory restrictions that limit them from processing massive volumes of data, making them unsuitable for processing large 3D imaging such as CT scans and MRI scans. Nevertheless, the combination of deep learning-based technique could be a possible solution for tackling the limitations of the aforementioned work.The SLAM [5] method is utilized to generate a dense point cloud map of the surroundings as well as estimate the position and orientation of the platform within the orchard. Some potential limitations of this method are mapping accuracy, expensive hardware requirements, and lacking generalization to other types of set up. Deep learning-based techniques such as CNN could be fruitful for certain tasks such as object detection and mapping optimization. Sandstone’s unusual microstructure affects both the mechanical characteristics of the rocks and how well hydrocarbons are stored and transported. Understanding the sandstone’s microstructure has benefited from scanning electron microscopy (SEM). However, it has been demonstrated that quantitative image analysis is challenging. Image segmentation is a crucial part of many AI systems for visual comprehension. Partitioning images into different segments or objects [6] is a key aspect of this process. Segmentation is crucial in many fields [7], including medical image analysis (such as tumor border extraction and measuring tissue volumes), autonomous vehicles (such as navigable surfaces and pedestrian identification), video surveillance, and augmented reality. From simple thresholding [8], region-growing [9], and more complex active contours [10], conditional and Markov random fields, lane marking detection [11], and sparsity-based [12] methods, the literature is replete with image segmentation algorithms. Recently, however, a paradigm shift has occurred as deep learning (DL) models have produced a new generation of image segmentation models with tremendous performance increases, often obtaining the highest accuracy rates on common benchmarks. The segmentation of images can be viewed as a multiclass classification issue with semantic labels for each pixel (semantic segmentation). In contrast to image classification, which predicts a single label for the entire image, semantic segmentation applies pixel-level labeling with a set of item categories (such as carbon, graphite, etc. in the lithium battery image) for all image pixels [13]. However, the computational complexity of the above-mentioned work is very huge. In this research, we proposed a unique approach to segment 3D volumetric objects. We evaluate how well computer vision, and specifically deep learning, can reduce the time spent analyzing SEM images from days to only a few seconds for a somewhat sized, high-resolution navigation area. For researchers in Digital Porous Media, Petroleum Science and Engineering, Water Science and Engineering, and Computational Fluid Dynamics who do not have the means to describe these rock samples themselves, this model could be useful to overcome the aforementioned challenges. The main contribution of the proposed model is the clarity of the approach, which includes dataset preprocessing based on the proposed model, generation of a 3D volumetric image from 2D slices, extraction of individual minerals from the 3D volume, extraction of each material individually based on pixel value, and finally the microstructural analysis. As a result, the suggested method has proven useful in capturing important information from porous materials. The initial experiment described was to determine if software could accurately capture microstructural changes across a range of shapes. We employed transfer learning on the VGG19 pretrained convolutional neural network (CNN) rather than gathering all the data required to train a network from scratch [14].VGG19 can learn an effective feature representation for the input image because it has been trained on a big dataset (ImageNet) to recognize a variety of objects and situations. To transfer learning to other tasks, such as image segmentation, similar pretraining can be used. The vast amount of parameters in VGG19 enable it to learn a detailed and intricate feature representation for the input image. When segmenting images, this is especially helpful because it allows for the correct segmentation of small objects and fine details. The 3D UNET model with VGG19, which relies on convolutional neural networks, is used for the bulk of the segmentation process because it has been shown to work well with multiclass microstructural data. The encoder–decoder architecture known as U-Net [15] was created especially for image segmentation applications. U-Net is highly suited for situations where small objects or features need to be precisely separated because of its ability to preserve delicate details. Due to its few parameters, U-Net is both computationally effective and simple to train. Based on specific requirements, such as the size of the dataset, computational resources, and further microstructural analysis tasks, our model proved to be effective for similar types of experiments. The study’s findings led to a categorization of several components that will improve future research. The significant innovations and insights of the work, in brief, are as follows:We have proposed a novel approach to solve the issue of volumetric image data segmentation which is very crucial to discover underground resources;The proposed approach deployed the concept of transfer learning along with VGG19 as the backbone, which makes the model unique from existing works in terms of performance;The findings and the results of our proposed model rely on the preprocessing of the training data. Our model is applicable to any type of image data such as FIB-SEM, CT, and MRI after proper annotation and mask generation for the training process;Our model extracts each particle individually based on their pixel value and represents their 3D volumetric visualization, which makes our work more unique than existing segmentation works. We have converted our volumetric data as multichannel objects, which is very crucial for further analysis of each region separately;Based on our model’s segmentation result, we have performed a microstructural analysis of each particle, which could be very useful for the measurement of individual particles in a mixed object. We have calculated the total area, average size, area percentage, etc. which makes our approach different from existing state-of-the-art work;Finally, in comparison with existing state-of-the-art works, our model shows impressive results in terms of accuracy.

## 2. Related Work

Because unusual reserves are inherently tight, it is essential to comprehend their microstructural characteristics, such as their mechanical properties, in order to effectively foresee how the formation will respond during the production and completion processes. Organic materials and petroleum both dwell in the micro- and nano-pores of unconventional reservoirs such as sandstones, shales, and coalbed methane formations. In order to quantify pore volume and explain pore structure, much recent research on these pores has relied on macroscopic, indirect measures [16]. Macroscopic porosity and pore structure can be measured and characterized indirectly by techniques such as nuclear magnetic resonance (NMR) spectroscopy, mercury injection capillary pressure (MICP) and surface area analysis. While these techniques are great for defining the pore structure, they do not actually create an image of the pore structure in great detail. Two- and three-dimensional pictures of nanometer-sized pores can be obtained using conventional scanning electron microscopy (SEM) or focused ion beam scanning electron microscopy (FIB-SEM) [17]. By firing an electron beam at a sample and then detecting the resulting signals, a two-dimensional image can be created using scanning electron microscopy (SEM). In FIB-SEM, an ion beam progressively mills away the surface as a series of sequential photos are captured to build a stack of images for a 3D representation, allowing for spot analysis in the mapping of elements across the surface. By combining these techniques, a 2D matrix representation of the sandstones’ minerals could be obtained. Reconstructing a 3D model of the matrix with FIB-SEM imaging allows us to see the microstructure and the connections between individual elements. There has been a lot of work based on CNN model object detection, image segmentation, etc. In order to quickly and accurately identify and classify various types of asphalt pavement cracks, Que et al. [18] proposed a method of automatic classification of asphalt pavement cracks utilizing a novel integrated GANs and improved VGG model. The suggested method is a two-step approach. To augment the dataset in the first stage and enhance the performance of the classification model, a GAN is employed to produce synthetic crack images. In the second stage, an improved VGG model is trained using the generated images. The enhanced VGG model comprises more convolutional layers that assist in extracting more characteristics from the images, increasing classification precision. However, the work has potential limitations such as require extensive computational resources, lack of generalization to real-world scenarios, and less diverse datasets. Yang Yu et al. [19] proposed a hybrid framework based on the CNN model and transfer learning to detect the cracks of various types of concrete. However, the crack detection system has a major disadvantage of considering whole patches as cracks, which could be resolved using a segmentation-based model. Noh et al. [20] presented an early article on semantic segmentation based on deconvolution. The encoder uses convolutional layers from the VGG 16-layer network, and the deconvolutional network takes the classification model as input and generates a map of pixel-wise class probabilities. Deconvolution and unpooling layers detect pixel-wise class labels and forecast segmentation masks. In SegNet, another promising study, Badrinarayanan et al. [21] suggested a convolutional encoder–decoder architecture for image segmentation. SegNet’s basic trainable segmentation mechanism comprises an encoder network that is topologically identical to the 13 convolutional layers of the VGG16 network, which is followed by a decoder network, and a pixel-wise classification layer. The decoder SegNet’s upsamples is unique. It uses max-pooling-step pooling indices for nonlinear upsampling of its lower-resolution input feature map(s). Upsampling is not required for learning. Using trainable filters to combine sparse upsampled maps and dense feature maps, SegNet has fewer trainable parameters than competing systems. Milletari et al. [22] proposed the V-Net, another prominent FCN-based model, for 3D medical picture segmentation. They used a dice coefficient-based goal function during model training to handle the substantial difference between foreground and background voxels. The network was trained only on prostate MRI data and can currently predict volume segmentation. A novel visual crack width measurement method based on backbone double-scale features for enhanced detection automation proposed by Tang et al. [23] is another significant contribution to the field of computer vision. The suggested method seeks to increase the precision and effectiveness of crack measurement and identification in diverse settings. The suggested method is computationally efficient in addition to having a high degree of accuracy because it only needs one forward pass of the backbone network to detect cracks and estimate their width. This qualifies it for real-time applications, including automatically spotting cracks in roads during inspections. It is crucial to note, though, that the proposed method has several drawbacks, such as limited crack patterns and generalization to real-world scenarios. A 2D CNN model with hyperparameter optimization performs well at predicting RC beams’ torsional strength [24]. However, we believe, for this type of work especially where we have to see the insights of the structure, the 3D-based CNN model outperforms the traditional 2D model. Progressive Dense V-net (PDV-Net) for the rapid and automatic segmentation of pulmonary lobes from chest CT images and the 3D-CNN encoder for lesion segmentation [25] are other notable medical image segmentation studies. Segmenting images from biological microscopy was made possible by U-Net, which was proposed by Ronneberger et al. [15]. Training and tuning their network on sparsely annotated images requires data augmentation. The U-design net consists of a context-capturing contracting path and a localization-enabling symmetric expanding path. In order to extract features, the down-sampling or contracting section uses an FCN-like architecture with 3x3 convolutions. When up-sampling or expanding, up-convolution (or deconvolution) is used to reduce the number of feature maps while increasing their size. To prevent losing pattern information, the down-sampling portion of the network’s feature maps is replicated in the up-sampling portion. A segmentation map is created by applying an 11 convolution to the feature maps, which labels each pixel in the input image with a predetermined label. The original U-Net design has undergone various suggested alterations and extensions in recent years to enhance its functionality. Modifications to the algorithm include adding skip connections between the encoder and decoder, employing residual blocks in the encoder and decoder, and using attention techniques to prioritize the features in the encoder and decoder. The U-Net architecture that Chen et al. [26] proposed may adaptively weight the feature maps based on their significance in the segmentation job. It has a channel and spatial attention mechanism. The use of U-Net in association with other deep learning models for image segmentation tasks has also been investigated by a number of experts. For instance, Li et al. [27] introduced the MMF-Net, which is a multi-scale and multi-model fusion U-Net that merges multiple U-Net models of various scales to increase segmentation accuracy. The performance of U-Net in image segmentation tasks can be considerably enhanced using transfer learning, which is a potent deep learning technique. In order to segregate the nuclei in a dataset of histopathology photos, Kong et al. [28] employed transfer learning to pre-train a U-Net on the ImageNet dataset. The capacity of the pre-trained model to learn high-level features that are essential for the segmentation task was cited by the authors as the reason why the pre-trained U-Net outperformed the U-Net trained from scratch.

## 3. Proposed Methodology

Our proposed model consists of dataset processing, model architecture, loss calculation, and preparing output for further analysis. Below, we have discussed each part separately.

### 3.1. Dataset Processing

We have used an open-source sandstone dataset for our experiment. The dataset contains a total of 448 images of size 512 × 512 × 512 of sandstone which contains multiple minerals. Every image in the volume is 256 × 256 × 256 in size. We make all images as the volume and represent them as a stack for 3D segmentation. Again, every original size image converts into multiple slices of size 256 × 256 × 256 based on system configuration. Every slice of the image is manually annotated to create a mask for training and testing purposes. Furthermore, the slices joint together and bring back to their original shape, and this procedure has been completed for every big image. The sandstone dataset conatins 4 particles in it. The black area which is pore (class 0) in the sandstone, the busy darkish area is mineral 1 (class 1), the grayish area which is most in this sample is mineral number 2 (class 2), and finally the bright or white region is mineral 3 (class 3) in Figure 1a,b as well as in Figure 1c,d. Figure 1e,f represents the cropped sample of the original image and corresponding mask. Table 1 represents the class and corresponding pixel value for each class.

### 3.2. Model Architecture

We have implemented the modified 3D UNET model for segmenting the 3D subvolume, which includes the stack of 2D images to speed up the segmentation result. Initially, our image slice is 256 × 256 × 256. To speed up the process and hardware requirements, we have converted the images in the batch of 64 × 64 × 64 using simple python code. Later on, those small chunks aggregated together to bring back to their original shape. This step has been completed for both training images and their corresponding masks. A total of 4 classes are defined for this segmentation, as our sample data contain 4 different materials: pore, mineral 1, mineral 2, and mineral 3. The data are further split into training data which is 80% of the total, while the validation data comprise 10% and the testing data comprise 10% of total. By adjusting the model after each epoch, a validation split enhances the model’s performance. Hyperparameter tuning is essential for improving the deep learning network’s performance, especially for challenging tasks such as the segmentation job. The hyperparameter tuning job includes the selection of activation function, optimizer, learning rate, batch sizes, number of epochs, etc. We have used dice loss and focal loss together to calculate the total loss of our model, which is proved to be better for overall accuracy estimation. Moreover, VGG19 is used as the backbone of the model. Our model used imagenet weight so that it could start from some good baseline and not from the scratch. Softmax is used as the activation function, as this a multiclass segmentation problem. We have used 0.0001 as the learning rate, which is good for initial training. Finally, we used Adam optimizer, as it is the best fit for our segmentation task. Table 2 describes the parameter settings of our model, which we have found to be the best for the accuracy. In Figure 2, our proposed architecture is built on using the concept of transfer learning. In the contraction part, we have added the pretrained model VGG19 so that we could start our training from a stable stage; then, it works in the manner of 3D UNET.

Figure 3a–c describes the prediction of our test image based on our model which shows almost accurate prediction. From the proposed architecture, we have used the slice of 64 × 64 × 64 from our initial 256 × 256 × 256 images as input. There are two paths in the complete model: one is contraction and another one is expansion. The difference between conventional neural network and U-NET is both paths are responsible for the concatenation of feature maps, which helps to achieve localized information. The proposed architecture is designed over existing 3D UNET architecture with VGG19 as the backbone. The conv layer for down-sampling is 3 × 3 × 3. Then, the max pooling we used is 2 × 2 × 2. After that, the up-sampling process starts, which is 2 × 2 × 2. Finally, we obtained one conv, which is 1 × 1 × 1: our output image. Initially, the model was trained and tested for a cropped size image, which is smaller than the original, and finally, the model is applied for the full volume of 512 × 512 × 512 images. For our further analysis, we have converted our output image into multichannel output. All segmented images are reconstructed and separated based on their pixel values. In our experiment, there are 4 segmentations based on 4 different types of materials. Finally, the images are converted to multidimensional images using exiting open source python software, which helps to separate each material as multichannel, which is very helpful to analyze the particles individually. Multichannel concepts convert the image into a binary image, which is further useful for analyzing each particle individually. Figure 3d,e represents the RGB and grayscale image of our segmented sample. In the color image, the red portion denotes the pore, while green is mineral 1, blue is mineral 2, and the white portion is mineral 3. Figure 3f, is the 3D representation of our segmented image, which we generate using free image acquisition and analysis software Zen lite. Furthermore, we have separated every individual particle based on their pixel value.

In Figure 4, we have represented the individual particle together for better visualization. Separating the individual particle is useful for further analysis. Figure 4a–h are the 2D visualization and their corresponding 3D visualization of each particle of the sample, which we have generated using free 3D image analysis software Zen lite. Figure 4i–k represent the particle together. This segmented volume was further used for our particle analysis work.

### 3.3. Loss Calculation

We have used the combination of dice loss and focal loss for our deep learning model. Multiclass Dice loss adjusts the weight of each class based on the square of label frequencies, similar to the original Dice loss. In the field of computer vision, the Dice coefficient is the metric for determining the degree of visual similarity between two images.

Focal loss applies the notion of focal loss to circumstances with low probabilities and high difficulty. When training for a job such as image segmentation, class imbalance can be an issue. For the purpose of concentrating training on challenging misclassified samples, focal loss modifies the cross-entropy loss as a whole. Focal loss can be calculated in “(1)”
(1)$$FL\left(pt\right)=-\alpha t{\left(1-pt\right)}^{\gamma }loglog\left(pt\right)$$
where *pt* is the extension to cross-entropy loss, alpha is the weighting factor and gamma is the tunable parameter. Small values of pt indicate that the loss is unaffected by the misclassification of the example. When pt is equal to 1, the factor becomes 0, and the loss for correctly categorized samples is reduced. The focusing parameter gamma allows for a gradual adjustment of the down-weighting of simple examples. If gamma is 0, then the focal loss is equivalent to cross-entropy. Then, the total loss is calculated using “(2)”.
(2)Total_loss=dice_loss+(1∗focal_loss)

### 3.4. Preparing Output for Particle Size Analysis

Our sample dataset contains different materials in it, including pores and materials. In this step, we have further analyzed our output to obtain the quantitative analysis of each mineral existing in the sample sandstone. After implementing the model to our original full volume samples, we have further segmented each class of sample images separately so that the analysis of particles could be completed easily. We have divided the predicted results into four segments: pore, mineral 1, mineral 2 and mineral 3.

Figure 5a–d represent the individual elements of our segmented sample based on their pixel value. First, we have segmented the full volume. Furthermore, we separated each particle. This is important for converting the sample in a binary image for particle analysis.

## 4. Particle Analysis

We have used the open source imagej-fiji tool to analyze our sanstone dataset. Each output voulme is converted into 8-bit binary images from which we have analyzed each particle individually. We have prepared our output volume as 4 classes which are segment 0 (pore), segment 1 (mineral 1), segment 2 (mineral 2), and segemnt 3 (mineral 3). Furthermore, we loaded each segment into imagej-fiji software. The pixel value is set to 0–3 as there are 4 different color contrasts for different materials. Furthermore, we have calculated the particle count, size of each particle, area, and average area for each individual segment.

Figure 6a–d are generated using open source imagej-fiji software, which represents only the binary bit of each segmented particle: either 0 or 1. Here, 0 represents the background or white in our figure, and 1 represents the existence of corresponding particles. Figure 7a–d represent the area and count of the particle. Furthermore, we have measured the particle count, area, average size and percentage individually in the whole volume. Table 3, Table 4, Table 5 and Table 6 represent the measurement of individual particles.

## 5. Result and Discussion

Our proposed model is applied to the sandstone dataset, and we have observed the performance of our model. We have observed that after a certain interval, increasing the number of epochs has a significant impact on loss, accuracy, F1 score and IOU. We have found the best result after running our model for 184 epochs which are described in Table 7. One important observation is that GPU computation is a must for this kind of experiment because CPU computation does not work or takes a long time due to the large volume of the computation.

To begin with, dice loss and focused loss were used to assess model performance as a function of epoch count. The objective here is to cut losses as much as possible. The graph clearly shows that after 20 epochs, the difference between training loss and validation loss begins to shrink. This suggests that after 20 epochs, the model begins to show signs of fitting the data. Hence, we observed the improvement of loss, accuracy, F1 score, and IOU value until 53. After that, the corresponding loss again started to increase and it continued until 80 epochs. After that, we again saw the downward trend of the statistics. So after having deep observation, we have come to know that the uptrend and downtrend continued in several intervals of the epochs. However, at one point, after 200 epochs, we have not observed any drastic improvement or changes in the performance. Therefore, the model was trained for 200 epochs, and the best value was found after running 184 epochs. Second, we looked at how the IOU score changed when the training and validation epoch counts changed. Visual inspection of the graph reveals a sharp rise in the IOU score for both training and validation between 50 and 190 epochs. Assuming 40 epochs have passed, the validation IOU score will begin to rise. This also means that after 40 epochs, the model’s performance begins to improve. We have trained the model for 200 epochs in total, which took 26.66 min in total, approximately 8 s per epoch using NVIDIA GPU P100 environment. We have observed that running this experiment on a CPU environment would take a couple of hours, even days, and most importantly, it might become stuck in the middle of the training. So, broadly speaking, our proposed method could skip unnecessary training time while working with datasets of similar size. Additionally, our proposed model could be extended to different kinds of volumetric image data such as CT and MRI. Talking about the limitation, our model’s quantitative metrics such as accuracy and IOU heavily rely on the proper preparation of training data. Annotation and masking for training is a very crucial part to obtain the best result. Again, the hybrid model’s effectiveness will be determined by the fineness of the training data and the particular 3D image segmentation job. To obtain an optimum result with real-world data using our proposed method, data preprocessing should be completed extensively prior to the experiment. Our proposed hybrid model might well be able to perform object segmentation tasks more accurately than either model alone by combining U-Net’s capacity to maintain small features with VGG19’s rich feature representation. Since U-Net is a pretty small model, it is known to be susceptible to overfitting when trained on a limited dataset. Our hybrid model’s use of VGG19’s learnt features has the potential to lessen overfitting and improve generalization performance. Our model is computationally efficient, as we applied the concept of transfer learning, which drastically reduced the number of parameters to be trained. So, based on the aforementioned advantages, our proposed model is best fit for segmentation tasks with a comparatively small dataset and where extracting individual features is important.

Table 8 shows that the proposed approach outperforms other existing strategies that take UNET-based deep learning algorithms into account.

Erdem et al. achieved an 86.61% F1 score using 2D UNET-VGG16 for aerial image segmentation [29]. According to Bahet et al., they achieved 73.76% IOU for the IDD dataset based on UNET with EfficientNetB7 [30]. A study by Nezla et al. [31] attempted to use semantic image segmentation to uncover hidden details for underwater images where they achieved an accuracy of 96.66% using UNET-based architecture. Nodirov et al. [32] proposed a 3D UNET based segmentation method which showed the F1 score of 88.53%. Albishri [33] proposed a 3D end-to-end UNET-based network for brain claustrum segmentation where they obtained an IOU of 70%. 3D Attention U-Net [34] explained the CADA-Aneurysm Segmentation Challenge where the final F1 score was 88.91%. The proposed model is more effective at segmenting the volumetric images than the current deep learning techniques, because the evolutionary result of the proposed method is superior. Moreover, most of the existing work explained the implementation of the model using 2D SEM images. However, our model has been implemented for volumetric 3D images which explained the use of 3D UNET for realistic three-dimensional datasets.

Figure 8a,b show the training loss and IOU and corresponding validation loss and IOU. These findings demonstrate the potential for using this method to properly identify mineral groups in SEM images, which has applications beyond just determining the presence of minerals or pores. As shown in Figure 8c,d, we have successfully identified the individual particle in our sandstone sample. From the result analysis of the total volume, we have concluded that almost 83% of our sample is mineral 2, while pore represents 9.11%, mineral 1 is 5.378%, and mineral 3 is 3.495%.

Table 9, describes the average size of each particle in our sample where it is found that mineral 2 is the largest in size in the whole volume of samples, which is 87603 (μm^2^), and pore, mineral 1, mineral 3 are 334.67 (μm^2^), 207.63 (μm^2^) and 324.81 (μm^2^).

## 6. Conclusions and Future Work

Testing the accuracy and IOU of image quality evaluation showed a remarkable 96.78% accuracy and 91.12% IOU. This model was shown to be efficient in classifying images as “excellent” or “poor” for further processing. Although our model was developed using the sandstone dataset for the goal of studying segmentation, it may be used and expanded for any type of SEM 3D volume image segmentation. Moreover, our proposed model was expected to be very useful for experiment with small datasets. To obtain initial insights into our experiment, our model could be a best initial start. Our model is less computationally complex based on the time taken for training. Talking about scalability, our model is scalable to other kinds of image segmentation with proper preprocessed data. Our model used the concept of transfer learning and includes the VGG19 model with 3D UNET, which shows a significant improvement in the result. The successful development of a machine learning method as an automated and robust feature extraction tool is crucial for recognizing porosity or particles in various porus materials. Size distribution could be accurately measured if the volume is segmented properly with an accurate deep learning model. Our model showed promising results and could be further implemented for any 3D volume image data. Considering the cost of X-ray computed tomography, our 3D volumetric SEM image segmentation technique could be the best alternative in terms of time and money. The suggested work has the potential to lead to the design of a deep learning-based segmentation model that is comparably less expensive in the future because the utilized algorithms are computationally less expensive and do not require considerable training. Microstructural analysis is one of the crucial tasks to detect the underground resources. The segmentation of individual particles or minerals of mixed material is a very important job for researchers in the field of porus media, chemical engineering, petroleum science, etc. to understand the insights of the materials. The proposed model is also effective for experimenting with small datasets to efficiently perform the segmentation and obtain the insights into the materials. Adding to the mentioned limitations in the discussion section, another potential limitation is that our hybrid model may not be generalizable for any types of 3D segmentation because of the preparation of the dataset. We plan to include the improvement of our model as a more general-purpose segmentation model. Additionally, the possible future extension of the model will include experimentation with more real-world data.

## Figures and Tables

**Figure 1 sensors-23-02640-f001:**
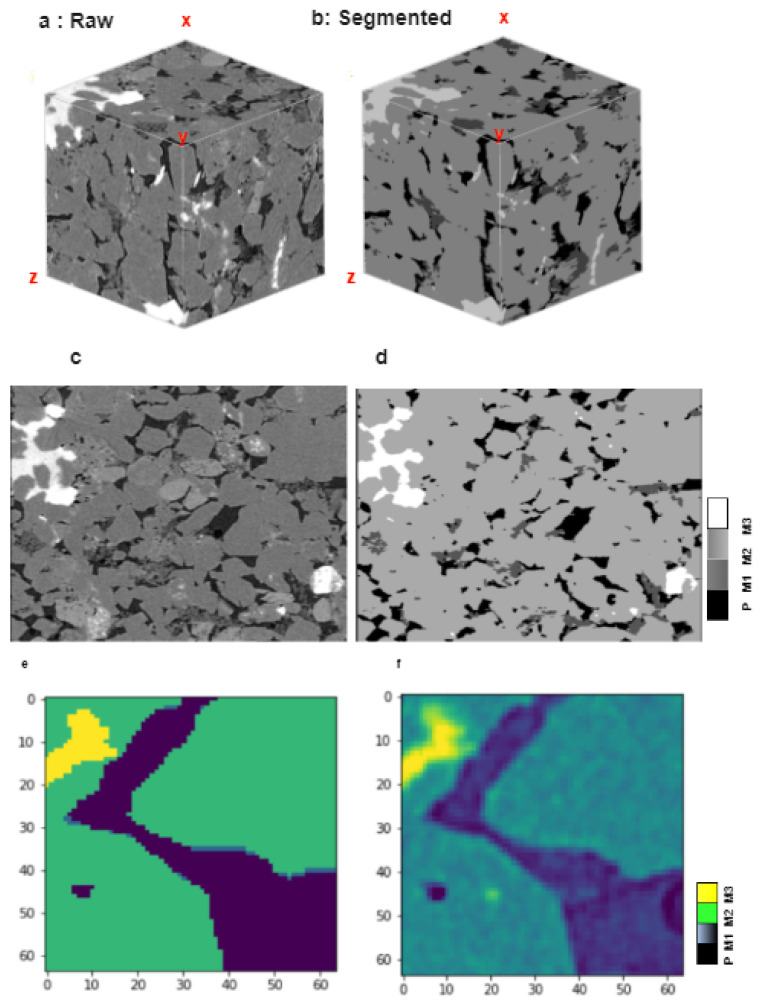
Represents the both 2D and 3D samples and train image/mask of individual slice. (**a**) Representation of 3D volume raw sample. (**b**) Representation of 3D volume segmented result. (**c**) Representation of 2D raw sample 512 × 512. (**d**) Representation of 2D segmented result. (**e**) 64 × 64 slice of training image. (**f**) 64 × 64 slice of training mask for model training.

**Figure 2 sensors-23-02640-f002:**
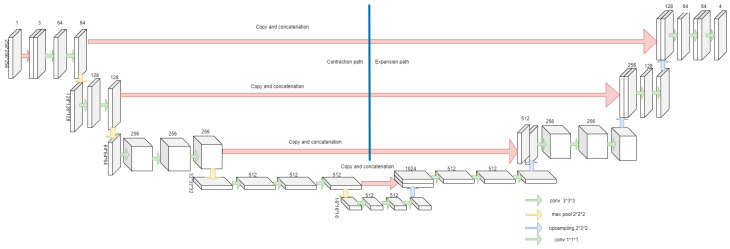
Architecture of our proposed model built combined with VGG19 AND 3D UNET.

**Figure 3 sensors-23-02640-f003:**
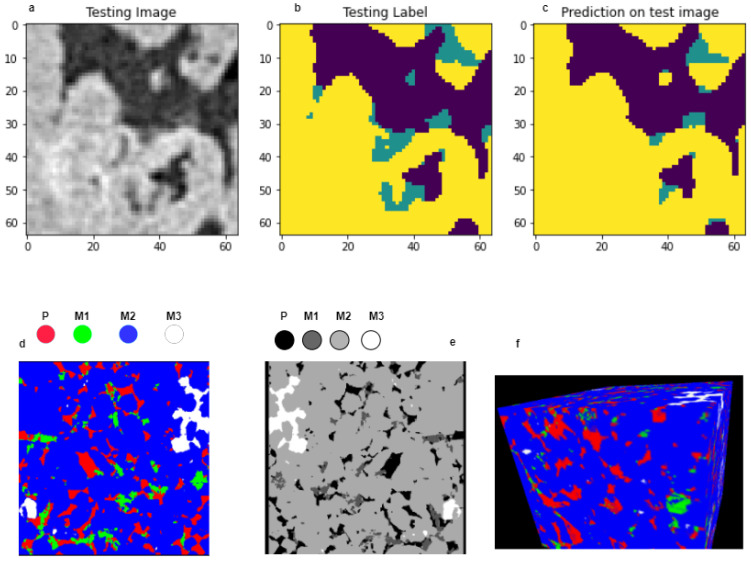
Represents prediction, segmented sample both in 2D and 3D. (**a**) Test image of individual slice. (**b**) Test mask of individual slice. (**c**) Prediction on test image. (**d**) Two-dimensional (2D) representation of segmented result in RGB where the red area represents pores, lime is mineral 1, blue is mineral 2, and finally, white represents mineral 3. (**e**) Representation of segmented 2D result in grayscale. (**f**) Representation of 3D volume segmented result.

**Figure 4 sensors-23-02640-f004:**
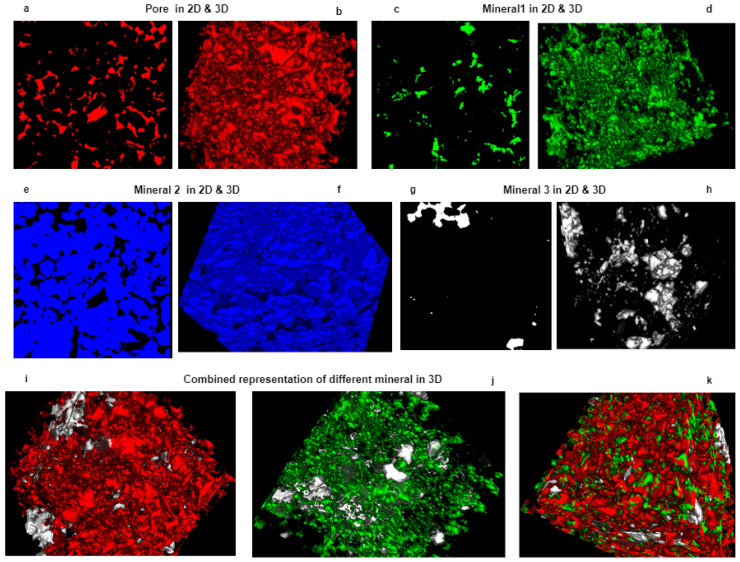
Representation of each particle individually. We have leveraged the free image analysis software Zen lite to represent our individual particle as multichannel data. (**a**,**b**) The red area represents the pore area of the sample for both 2D and 3D. (**c**,**d**) The lime area represents the mineral 1 area of the sample for both 2D and 3D. (**e**,**f**) The blue area represents the mineral 2 area of the sample for both 2D and 3D. (**g**,**h**) The white area represents the mineral 3 area of the sample for both 2D and 3D. (**i**–**k**) Three-dimensional (3D) representation of combined particles.

**Figure 5 sensors-23-02640-f005:**
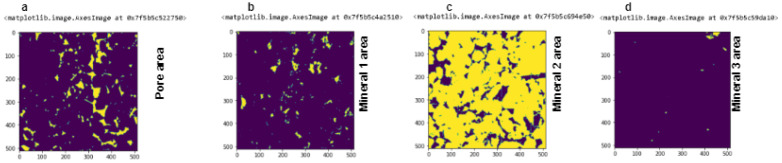
Individual particles of segmented images based on the pixel value on particular slices. (**a**) Represents pore area. (**b**) Represents mineral 1 area. (**c**) Represents mineral 2 area. (**d**) Represents mineral 3 area.

**Figure 6 sensors-23-02640-f006:**
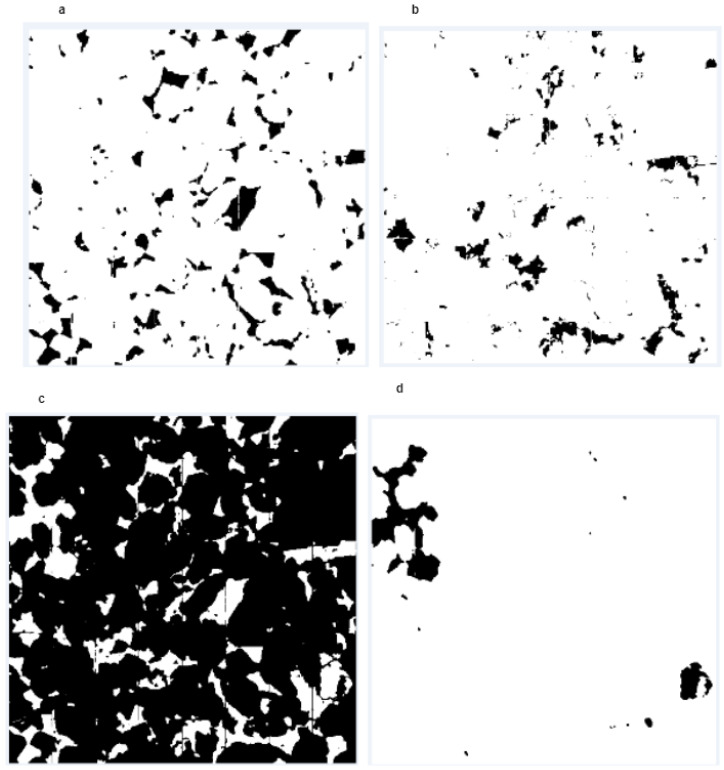
(**a**,**b**) Eight (8)-bit binary conversion of our individual segmentation (pore and mineral 1. (**c**,**d**) Eight (8)-bit binary conversion of our individual segmentation (mineral 2 and mineral 3).

**Figure 7 sensors-23-02640-f007:**
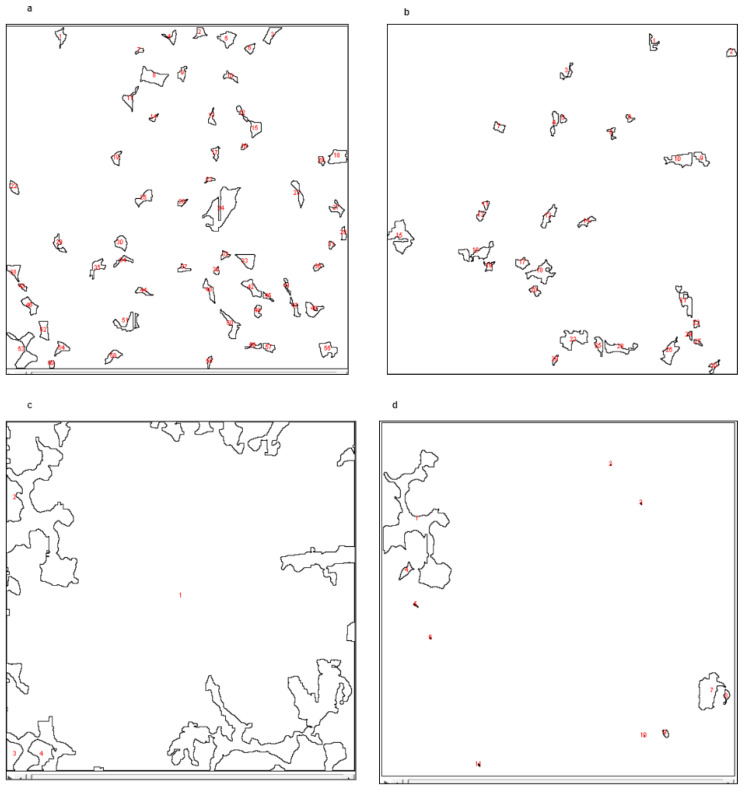
(**a**,**b**) Separation of particle and count individually (pore and mineral 1. (**c**,**d**) Separation of particle and count individually (mineral 2 and mineral 3).

**Figure 8 sensors-23-02640-f008:**
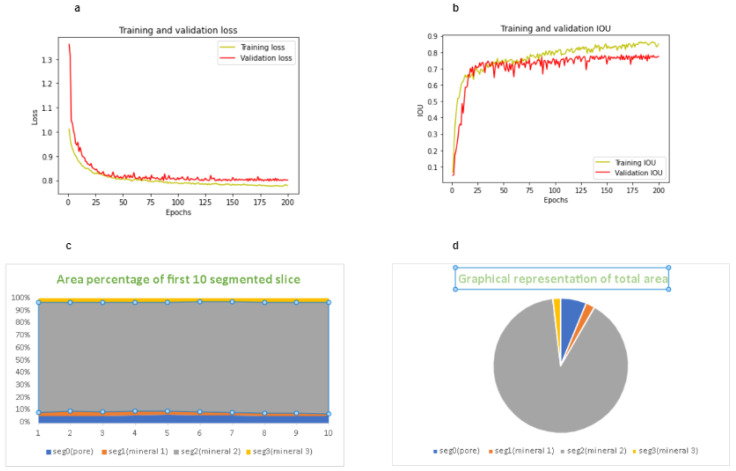
(**a**) Loss during training and testing as a result of the number of epochs. (**b**) IOU during training and testing as a result of the number of epochs. (**c**) Percentage of average area for each particle for the first 10 samples. (**d**) Representing the area of each particle individually.

**Table 1 sensors-23-02640-t001:** Details of our dataset labeling.

Serial No.	Class Name	Value of Pixel
1	Pore	0
2	Mineral 1	1
3	Mineral 2	2
4	Mineral 3	3

**Table 2 sensors-23-02640-t002:** Description of the parameters for our model and their corresponding values.

Serial No.	Parameters	Corresponding Values
1	Encoder weights	Imagenet
2	Backbone	VGG19
3	Activation function	Softmax
4	Loss	Dice loss and Focal loss
5	Patch sizes	64
6	Num of classes	4
7	Channels	3
8	Learning rate	0.0001
9	Optimizer	Adam

**Table 3 sensors-23-02640-t003:** Size analysis of segment 0 (pore) in the segmented output.

Slice	Count	Total Area (μm^2^)	Average Size (μm^2^)	Area Percentage%
1	60	13,513	225.217	5.155
2	60	14,333	238.883	5.468
3	59	14,798	250.814	5.645
4	58	15,730	271.207	6.001
5	52	16,308	313.615	6.221
6	53	16,304	307.623	6.219
7	49	15,946	325.429	6.083
8	49	15,210	310.408	5.802
9	47	15,024	319.66	5.731
10	47	14,070	299.362	5.367

**Table 4 sensors-23-02640-t004:** Size analysis of segment 1 (mineral 1) in the segmented output.

Slice	Count	Total Area (μm^2^)	Average Size (μm^2^)	Area Percentage%
1	30	6892	229.733	2.629
2	33	9161	277.606	3.495
3	27	8761	324.481	3.342
4	25	8833	353.32	3.37
5	29	7409	255.483	2.826
6	33	7087	214.758	2.703
7	31	6319	203.839	2.411
8	28	5521	197.179	2.106
9	30	5291	176.367	2.018
10	29	5050	174.138	1.926

**Table 5 sensors-23-02640-t005:** Size analysis of segment 2 (mineral 2) in the segmented output.

Slice	Count	Total Area (μm^2^)	Average Size (μm^2^)	Area Percentage%
1	4	228,737	57,184.25	87.256
2	3	227,552	75,850.667	86.804
3	5	229,053	45,810.6	87.377
4	4	227,013	56,753.25	86.599
5	3	227,379	75,793	86.738
6	2	232,841	116,420.5	88.822
7	3	234,461	78,153.667	89.44
8	4	240,228	60,057	91.64
9	4	240,143	60,035.75	91.607
10	3	240,652	80,217.333	91.801

**Table 6 sensors-23-02640-t006:** Size analysis of segment 3 (mineral 3) in the segmented output.

Slice	Count	Total Area (μm^2^)	Average Size (μm^2^)	Area Percentage%
1	11	7411	673.727	2.827
2	12	7694	641.167	2.935
3	11	7723	702.091	2.946
4	11	7741	703.727	2.953
5	11	7527	684.273	2.871
6	11	7341	667.364	2.8
7	12	7439	619.917	2.838
8	13	7655	588.846	2.92
9	12	7868	655.667	3.001
10	14	7961	568.643	3.037

**Table 7 sensors-23-02640-t007:** Fit the model with different epoch values and analyze the performance.

Epoch	Loss	Accuracy	F1 Score	IOU
20	77.29	92.12	93.21	87.74
40	77.16	92.56	93.41	88.05
53	76.88	91.88	94.45	89.77
60	77.20	93.47	93.40	88.04
80	77.19	93.59	92.87	87.25
100	76.84	94.19	94.48	89.82
150	76.73	95.28	94.82	90.38
**184**	**76.60**	**96.78**	**95.25**	**91.12**
197	77.06	91.01	93.35	88.02

**Table 8 sensors-23-02640-t008:** Performance result comparison with different existing methods.

Model	Accuracy	F1 Score	IOU
2D UNET- VGG16 [29]	–	86.61	–
UNet with EfficientNetB7 Encoder [30]	–	–	73.76
UNET for underwater images [31]	96.66	–	–
3D U-Net-BrainTumor image [32]	–	88.53	–
AM-UNet [33]	–	–	70
3D Attention U-Net [34]	–	88.91	–
**Our Method**	**96.78**	**95.25**	**91.12**

**Table 9 sensors-23-02640-t009:** Average size of each particle in total volume.

Serial No.	Particle Class	Average Size (μm^2^)
1	seg0 (pore)	334.6775089
2	seg1 (mineral 1)	207.632
3	seg2 (mineral 2)	87,603.9935
4	seg3 (mineral 3)	324.8126

## Data Availability

Not applicable.

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
