# Peer review of "Deep Learning-Based Segmentation of 3D Volumetric Image and Microstructural Analysis"

_sensors, 2023, doi:10.3390/s23052640_

Round 1
Reviewer 1 Report
This manuscript proposed a deep learning-based method for 3D volumetric image and microstructure analysis, where convolutional neural network (CNN) was employed for the task of interest. In the proposed method, a 3D UNET based on CNN was established together with VGG19 to conductmulticlass segmentation of publicly available sandstone 11 datasets to analyze their microstructures using image data based on four different objects in the 12 samples of volumetric data. The performance of the proposed method has been validated using 442 2D image samples, with satisfactory results. Overall, the topic of this research is interesting, and the manuscript was well organised and written. The detailed comments are provided as follows.
1. The contributions and innovation of the paper should be clearly clarified in abstract and introduction.
2. Broaden and update literature review on CNN/deep networks and its application in image processing and data analysis. E.g. Vision-based concrete crack detection using a hybrid framework considering noise effect; Torsional capacity evaluation of RC beams using an improved bird swarm algorithm optimised 2D convolutional neural network.
3. The performance of the CNN model is heavily dependent on the setting of hyperparameters. How did the authors optimise the network parameters to achieve the best classification accuracy?
4. More information should be added on how the authors set training, validation and test samples.
5. Training time should be also considered as one of metrics for performance evaluation.
6. The conclusion part should be extended and more future research can be added to this part.
Reviewer 2 Report
Title: Deep Learning-based Segmentation of 3D Volumetric Image and Microstructural Analysis Major revision
The manuscript introduces a deep learning-based segmentation approach for 3D volumetric images and microstructural analysis. The research is intriguing and offers valuable results, but the current document has several weaknesses that must be addressed in order to produce a final product that is worthy of publication.
(1) At a thematic level, the proposal presents an interesting vision, as the segmentation of 3D volumetric images and microstructural analysis would be a useful resource for engineers. However, a comprehensive understanding of 3D segmentation is not limited only to accuracy and IOU. This limitation is an important aspect of the proposal, which should be assumed with more rigor and realism in the development of the manuscript's argumentation.
(2) The introduction should present a more comprehensive view of the problems related to the topic, with citations to state-of-the-art references, For instance, 3D global mapping of large-scale unstructured orchard integrating eye-in-hand stereo vision and SLAM, Computers and Electronics in Agriculture.
(3) The document contains a total of 26 references, which is not enough to support the research. Additionally, the research gap is not well presented.
(4) The article combines Unet and Vgg19 for object segmentation, which is innovative and holds promise. However, at the end of the article, when analyzing the proposed method, it lacks corresponding indicators and cannot fully demonstrate the advantages of the method.
(5) On a general level, the proposed segmentation technology is reasonable and the explanation of the work's objectives is valid. However, the limitations of the work are not rigorously assumed and justified.
(6) In the related work section, the contribution of your work should be highlighted by placing it in the context of previous work in the same domain.
(7)For VGG application, references keywords such as integrated generative adversarial networks and improved VGG model, should be mentioned. For Unet segmentation application, references keywords such as backbone double-scale features for improved detection automation, should be mentioned.
(8) The advantages of Vgg19 should be described in more detail.
(9) The quality of the figures should be improved. The marks on Figures 1 and 3 should be placed in the appropriate position, and the figures should be redrawn.
(10) Based on the description in the second paragraph of Chapter 3, Figure 6 does not effectively demonstrate the effect. Please resubmit a picture of the particle counting area calculation.
(11) Please modify Figures a and b of Figure 7.
(12) On line 265, Table 8 should be changed to Table 9. Please carefully check the correspondence of each table and figure number.
(13) To more comprehensively prove the feasibility of the method, you can add additional indicators such as the size and computation amount of the model.
(14) In the conclusion section, please objectively describe your experimental results.
(15) The method in the paper lacks explanation of limitations.
(16) I recommend including a discussion of the limitations of the model size consideration in this review in the limitations assessment. This part of the document can be improved and completed with more rigor.
Round 2
Reviewer 2 Report
The majority of the comments have been effectively addressed by the authors, with only one minor correction remaining. Comment (7) has been misinterpreted. Rather than modifying the keywords for this manuscript, the authors should focus on discussing the suggested two references.
Author Response
Reviewer comment: The majority of the comments have been effectively addressed by the authors, with only one minor correction remaining. Comment (7) has been misinterpreted. Rather than modifying the keywords for this manuscript, the authors should focus on discussing the suggested two references.
Author's response: Thank you for your comment. We have modified our revised version as per the reviewer comment. The changes can be found from line number 137-145 and 165-174 in our revised manuscript.